# Colorectal Cancer Fast Tracks: Cancer Yield and the Predictive Value of Entry Criteria

**DOI:** 10.3390/cancers15194778

**Published:** 2023-09-28

**Authors:** Linnea Uebel, Indy Kromodikoro, Nils Nyhlin, Michiel van Nieuwenhoven

**Affiliations:** 1Department of Internal Medicine, Division of Gastroenterology, Örebro University Hospital, Region Örebro County, SE 70116 Örebro, Sweden; linnea.uebel@regionorebrolan.se (L.U.);; 2Department of Internal Medicine, Division of Gastroenterology, Faculty of Medicine and Health, Örebro University, SE 70182 Örebro, Sweden

**Keywords:** colorectal carcinoma, colonoscopy, screening, epidemiology, symptoms

## Abstract

**Simple Summary:**

Fast-track pathways to detect colorectal cancer have been implemented in several European countries, using different entry criteria. We analyzed 2539 fast-track colonoscopies including referrals in detail, in order to calculate the predictive values and odds ratios of different red flags with respect to the risk for colorectal cancer. Interestingly, we observed that a number of red flags were not at all associated with an increased risk for colorectal cancer. On the other hand, the variable with the highest predictive value: a positive fecal occult blood test (FOBT), is not always part of colorectal cancer fast tracks. These findings are important when selecting patients for fast-track colonoscopy. We propose that the entry criteria should be limited to the following signs/symptoms: patients > 40 years with one or more of the following signs/symptoms: positive FOBT, abnormal radiology, abnormal rectal examination, visible blood in stool, in the absence of hemorrhoids and unexplained iron-deficiency anemia.

**Abstract:**

Background: Fast-track pathways for diagnosing colorectal cancer (CRC) have been implemented in several European countries. In Sweden, a substantial number of CRC are diagnosed via the Swedish Standardized Course of Care for colorectal cancer (SCC-CRC). We evaluated the SCC-CRC in terms of CRC yield, and predictive values and odds ratios (OR) for the entry criteria. Methods: We retrospectively analyzed all 2539 patients referred for SCC-CRC colonoscopy between September 2016 and December 2020. Entry criteria and colonoscopy outcomes were analyzed. Results: CRC yield was 16.4%. Highest positive predictive values (PPVs) were seen for abnormal radiology (PPV 30.5%, OR 4.7 (95% CI 3.4–6.4) *p* < 0.001), abnormal rectal examination (PPV 28%, OR 3.6 (95% CI 2.7–4.8) *p* < 0.001), and anemia (PPV 24.8%, OR 2.2 (95% CI 1.5–3.1) *p* < 0.001). Some entry criteria showed no significant risk increase, i.e., visible blood in stool/rectal bleeding, change in bowel habits, and the combination of changed bowel habits plus anemia. A positive fecal immunochemical test (FIT), although not part of the SCC-CRC, showed the highest OR: 9.9 (95% CI 4.5–21.7) *p* < 0.001) and PPV of 18.8%. Conclusions: CRC yield from the SCC-CRC is slightly higher compared to other European fast tracks. A number of entry criteria showed no benefit towards assessing CRC risk. FIT testing should be included in CRC fast tracks to increase diagnostic efficacy.

## 1. Introduction

In Sweden, fast-track pathways for almost all types of cancer have been developed and implemented in recent years. 

Besides Sweden, there are a number of European countries that have established fast-track pathways for patients with symptoms suggestive of colorectal cancer. Fast tracks were developed to improve patient outcomes by reducing the time between suspicion, diagnosis, and treatment initiation, ultimately enhancing the chances of successful cancer management and recovery. Examples of these are the two-week wait (2WW) in the UK [1], priority class “Code B” for non-delayable endoscopy in Italy [2], “pakkeforlop” in Norway [3], “pakkeforløb” in Denmark [4], and Standardized Course of Care for colorectal cancer, SCC-CRC, in Sweden [5]. The criteria for the SCC-CRC were formulated by the Regional Cancer Center by a dedicated working group. This group includes representatives from all parts of the patient's health care chain, such as general practitioners, surgeons, oncologists, contact nurses, pathologists, radiologists, and one or more patient representatives. To the authors´ knowledge, there is no nationwide fast track in Spain, but Catalonia and Valencia have implemented their own fast-track programs [6,7].

European fast tracks display similar entry criteria, with some differences as to the extent and detail of the criteria (Table 1). The most common is visible blood in stool/rectal bleeding, followed by anemia, altered bowel habits, abnormal finding on rectal examination or rectoscopy, weight loss, abdominal pain, suspect radiological finding, and finally, palpable abdominal mass. Age is included in some form in most of the fast tracks. Patient sex is not generally taken into account, despite the fact that male sex is a known risk factor [8]. For some fast tracks, the criteria are few and unconditional, for example, in Italy [2], or are conditioned only by a common minimum age for all criteria, for example, in Norway and Denmark [3,4], whereas, in the UK, most of the individual criteria are associated with a specific minimum age, ranging from above 40 to 60 years, depending on the criterion [1]. In Sweden, hemorrhoids, which may cause visible blood, should first be treated adequately before a SCC-CRC referral for a colonoscopy. There is a difference regarding fecal occult blood test/fecal immunochemical testing (FOBT/FIT) as part of the fast track, where in some fast tracks, it is regarded as non-obligatory (UK and Valencia, Spain), or testing is suggested if the person does not meet the required age limit (UK), while others do not include it (Sweden, Norway, Denmark, and Italy). The established maximum waiting time for assessment by a specialist differs from 10–21 days, or 30 days from a suspected malignancy to the start of treatment in Catalonia, Spain [1,2,3,4,5,6,7].

However, several studies have reported that the use of fast tracks does not increase cancer detection rate or cancer outcome, with a higher burden on endoscopic services, leading to a displacement effect affecting routine patients [9,10,11]. In the UK, the number of fast-track referrals has increased steadily since the implementation of the 2WW in 2000, while no improvement in CRC yield has been noted [10]. Several studies report similar findings, with a CRC yield of 3.1 to 7.7% [9,10,11]. An Italian study reported a CRC yield of 8% in 2019 and 7.7% in 2020; however, these numbers also include patients in need of urgent endoscopy within 72 hours [2]. A Danish study showed that non-fast-track cancer patients had to wait longer for their diagnosis than before fast tracks were implemented [12]. In our region, we did not observe any benefit regarding prognostic outcome of CRC for patients investigated via the fast-track vs. standard pathway [13]. 

In Sweden, the first version of the SCC-CRC, implemented in 2016, was considered sub-optimal in terms of the entry criteria, resulting in high numbers of referrals and difficulties to adhere to the 10-day waiting period. Therefore, a revision of SCC-CRC was implemented in 2019. The revised entry criteria for colonoscopy as part of the SCC-CRC comprise visible blood in stool without obvious source of bleeding upon rectoscopy, or persistent blood in stool despite adequate treatment of a probable source of bleeding for > 4 weeks (hereafter called “visible blood in stool”), anemia due to blood loss, abnormal rectal examination, either manual or via rectoscopy, abnormal radiology such as wall thickening or a suspected mass, abnormal tissue diagnostics, and finally, a combination of altered bowel habits plus anemia or rectal bleeding. A patient fulfilling at least one criterion should be examined by colonoscopy within 10 days. The most notable change involved the symptom of altered bowel habits such as onset of diarrhea or constipation, which previously was an entry criterion but was later considered relevant only in combination with anemia or rectal bleeding [5]. During the first year after the revision, almost 20,000 patients in Sweden were examined via the SCC-CRC; however, less than half within the intentioned 10 days [14]. In addition, we previously showed that 37.5% of all CRC-cases diagnosed through colonoscopy in our region were diagnosed via a routine waiting list [13]. 

## 2. Materials and Methods

We aimed to evaluate the cancer yield and predictive values of entry criteria in patients referred according to the Swedish SCC-CRC.

### 2.1. Study Design and Patient Material

We performed a retrospective review of medical records from all patients accepted for a SCC-CRC colonoscopy within the 3 hospitals of the Region Örebro County from September 2016 to December 2020. In total, 2790 patients were accepted for colonoscopy during this period. 

The colonoscopies were performed by experienced gastroenterologists or colorectal surgeons in an approximate 1/1 ratio. The majority of fast-track referrals came from general practitioners (GPs) and a minority were hospital-based referrals. All GPs and hospital specialists have knowledge about and access to the fast-track entry criteria. Subsequently, all referrals were reviewed centrally by experienced gastroenterologists from the endoscopy unit at the university hospital Örebro, and they decided about the acceptance of the referral. Our study population consists of two groups due to the fact that the entry criteria for the SCC-CRC were revised during the study period. The criterion “altered bowel habits for > 4 weeks in a patient > 40 years” was changed to “altered bowel habits combined with anemia or visible blood in the stool”. This criterion was recently removed after an update of the SCC-CRC in 2022, which was after the data collection for this study was completed. Instead, a new criterion constitutes “altered bowel habits in patients > 40 years with a positive FIT”. 

### 2.2. Data Collection

For each individual patient, we extracted all data manually from medical records and collected information about reasons for referral (SCC-CRC entry criteria), demographics, laboratory values if available, and colonoscopy outcome. Symptoms not mentioned in the referral were assumed to be absent. Cases of synchronous CRC were registered as one case per patient. All pathological findings were included. High-risk adenomas were defined as high-grade dysplasia, adenomas ≥ 10 mm, multiple adenomas (≥3), villous histology, or serrated polyps with dysplasia or ≥ 10 mm in size. For laboratory values, the cut-off period was 1 month before investigation and the most recent value was used, except for FIT, which was considered positive if at least one of three consecutive tests was positive. The study was approved by the Regional Ethical Reviews Board in Uppsala (Dnr. 2019-00271).

### 2.3. Statistical Analysis

We used IBM SPSS Statistics version 27.0. We did Pearson’s Chi^2^ testing and present absolute numbers and percentages for incidence, positive predictive values (PPV), negative predictive values (NPV), sensitivity, specificity, and odds ratio (OR). For the analysis of OR for different symptoms/findings, the comparator was patients without that symptom. Logistic regression was used to adjust for age, sex, and other criteria, when examining risk. Non-normally distributed numerical variables were analyzed by Mann–Whitney’s U test and presented using median and interquartile range (IQR). Statistical significance was set at *p* < 0.05. For the pooled analysis, all criteria that were present at some time between 2016–2020 were used, despite the fact that some only existed in one of the two groups. Since no new symptom was either introduced or completely removed in the 2019 update, and only different combinations of the same criteria were used, we also analyzed the presence of the old SCC-CRC entry criteria in the second group, and vice versa.

## 3. Results

### 3.1. Patient Inclusion and Characteristics

In total, 2790 patients were referred and accepted for colonoscopy according to the SCC-CRC. Of these, 251 (9.0%) were excluded. The inclusion process is shown in Figure 1. In total, we included 2539 patients. Of these, 1271 were investigated before the implementation of the revised SCC-CRC criteria in 2019. Baseline characteristics are presented in Table 2.

### 3.2. Colonoscopy Outcome

The colonoscopy outcomes are displayed in Figure 2. The pooled colorectal cancer yield was 16.4%. Left-sided cancers (descending colon and rectum) were the most common (63.0% of all CRC findings). The prevalence for adenomas with high-grade dysplasia was 92 (3.6%). Fourteen patients had both CRC and adenoma with high-grade dysplasia, and sixty-one patients had both CRC and high-risk adenomas. The combined prevalence of CRC and/or adenoma with high-grade dysplasia (patients with both findings only accounted for once) was 494 (19.5%). The respective prevalence of CRC and/or high-risk adenomas was 740 (29.1%).

### 3.3. Performance of SCC-CRC Criteria

The PPVs and ORs of the entry criteria are displayed in Table 3. The highest PPVs and ORs for the pooled data were seen for abnormal radiology, abnormal rectal examination, and anemia. Visible blood in the stool and altered bowel habits did not show any significantly increased risk for CRC. The combination of altered bowel habits and anemia showed a PPV of 26.2% and a statistically significant risk increase for CRC in the pooled data, with similar findings from the 2019–2020 period. When the altered bowel habits criterion was combined with rectal bleeding, the PPVs and OR did not show an increased risk for CRC.

### 3.4. Adjusting for Confounders

Logistic regression adjusting for age and sex showed increased ORs for most criteria (Table 4). For the pooled analysis, the adjusted OR for anemia, visible blood in stool, and change in bowel habits remained roughly the same, whereas the adjusted OR was increased for abnormal radiology, abnormal rectal finding, and the combination of changed bowel habits and blood in stool. When adjusting for FIT in the pooled regression analysis, the ORs of most other criteria in the analysis increased, and the adjusted OR for FIT was 9.9 (CI 95% 4.5–21.7) *p* < 0.001). Also, it improved the ORs for most other criteria.

Results from logistic regression adjusting for age and sex, presented for the two study populations (2016–2018 and 2019–2020) and the pooled data. OR = odds ratio. Missing cases excluded from analysis. Statistics: Nagelkerke R^2^ for the 2016–2018 analysis, 2019–2020 analysis, and 2016–2020 pooled analysis: 0.160, 1.53, and 0.159, respectively. Missing cases 14.2%, 13.0%, and 13.6%, respectively.

### 3.5. FIT: The New Criterion

We also analyzed the new 2022 criterion: “changed bowel habits combined with a positive FIT in a patient > 40 years of age”. This was present in 86 (45.7%) of the CRC group and 393 (34.9%) of the non-cancer group, showing a PPV of 18.0%, NPV of 87.8%, sensitivity of 45.7%, specificity of 65.1%, and unadjusted OR of 1.6 (95% CI 1.2–2.1), *p*-value 0.004). When adjusting for age and sex in a logistic regression model with the other criteria, the adjusted OR was 2.2 (95% CI 1.5–3.1), *p* < 0.001. 

### 3.6. Laboratory Values

Although not included in the SCC-CRC, from 1314 patients in the pooled data (51.8%), a FIT was obtained. Hb values were significantly lower in CRC patients (Table 5).

## 4. Discussion

The cancer yield in the total SCC-CRC cohort was 16.4%, which was the fourth most common finding. The most common endoscopic findings were diverticulosis, adenomas with low-grade dysplasia, and hemorrhoids. These findings often overlap with CRC presentation. The high prevalence of diverticulosis and adenomas could be expected, since these are common findings in an elderly population. The high adenoma detection rate, which is a major quality parameter for colonoscopy, demonstrates that high-quality colonoscopies were performed in this study. During the study period, the criteria for high-risk adenomas and their follow-up were adjusted [15]. In this retrospective study, we used the old criteria. 

### 4.1. Cancer Yield

Similar European fast tracks report lower cancer yield: 3.1% to 7.7% for the 2WW in the UK [9,10,11] and 7.7% to 8% in Italy [2]. Higher cancer yields have been reported by studies from the Spanish regions of Valencia and Catalonia, ranging from 15.6% to 28.7% [7,16,17]. It should be noted, however, that the prevalence numbers reported in the latter study also includes cases of CRC found with CRC screening [17]. The differences in CRC yield between the different countries may be explained by the fact that, in our study population, and in the Spanish studies, all fast-track referrals were reviewed by a gastroenterologist before being accepted. This review allows for the assessment of the quality of the referral, but also to exclude patients who recently underwent a colonoscopy or CT colonography with insignificant findings. There are no numbers available regarding CRC yield from Norwegian or Danish fast tracks.

### 4.2. Predictive Values of Fast-Track Criteria

Although the 2016–2019 and the 2019–2020 cohorts are not completely identical, the differences were small and, for this reason, we present our data in both separate cohorts and a pooled cohort. Three of the seven Swedish fast-track criteria showed a significant PPV and OR for CRC in all analyses: anemia, abnormal rectal examination, and abnormal radiology. Altered bowel habits in combination with blood in stool demonstrated an increased risk only in the pooled analysis when adjusting for age and sex. Furthermore, two criteria were not associated with an increased risk: visible blood in stool and altered bowel habits combined with anemia. A negative association for CRC risk was observed for altered bowel habits as a solitary symptom. 

Abnormal radiology finding showed the highest PPV and OR in both study groups and in the pooled cohort. Nevertheless, this criterion is only included in CRC fast tracks in Sweden and Italy [5,17]. However, this criterion is of limited clinical use in the context of assessing the need for fast-track colonoscopy, since it would be impractical to routinely perform radiological imaging before colonoscopy.

An abnormal rectal examination is an entry criterion for CRC fast tracks in Sweden, UK, Norway, Valencia (Spain), and Catalonia (Spain) [5,6,7]. GPs in Sweden are required to perform a rectoscopy prior to an SCC-CRC referral to rule out benign reasons for symptoms, usually visible blood in the stool. 

Anemia is an entry criterion for CRC fast tracks in Sweden, Denmark, UK, Italy, and regionally in Spain, with some differences to age cut-offs, type of anemia, and combination with other symptoms. In our study, bleeding anemia showed a PPV of 24.8% and an adjusted OR of 2.1 for the pooled data. Other studies show a PPV of 9.9% [18]. However, bleeding anemia also seems to be a common reason for referral for a routine colonoscopy when it is presented as the only symptom, most likely affecting its predictive value in CRC fast tracks [13]. In our cohort, CRC patients had a lower median Hb value compared to non-CRC patients, which is in agreement with previous studies [18,19,20]. The pooled OR of 2.1 is not very high, since anemia may have other causes.

Altered bowel habits are a prompt for urgent assessment for CRC according to several European fast-track pathways. Our results demonstrate that altered bowel habits are not at all associated with an increased risk for CRC. Other studies confirm the low predictive value of altered bowel habits with respect to CRC [21,22,23,24]. 

Visible blood in stool/rectal bleeding, although generally considered as a red flag, did not show any increased risk for CRC. The European fast tracks seem to agree upon its inclusion as a criterion, with some differences to age cut-offs and combination with other symptoms. A likely explanation for its low predictive value is the ambiguity of this criterion. Many referrals mentioned that other bleeding sources could not be ruled out, despite the presence of hemorrhoids. These differences in the interpretation and the uncertainty of the referring doctor likely affect its predictive value. Previous studies show a PPV ranging from 0.6% in the general public [24], to 6.6–21.2% in patients aged 60 years or older [22], to a pooled estimate of 8.1% [18], suggesting that visible blood alone is no reason for fast-track investigation [25].

Testing for FIT is included in the fast-track pathways in the UK, Valencia, Spain, and Italy. It was not part of the SCC-CRC during the time of this study. In April 2022, the SCC-CRC criteria were updated again and now includes positive FIT in patients > 40 years combined with changed bowel habits, but this is still not a compulsory part of the SCC-CRC as a whole. In our study, the PPV of a positive FIT was 18.8%, which is slightly higher than in other studies [26], and a positive FIT showed the highest OR of all analyzed symptoms and findings in this cohort with an unadjusted OR of 11.4. Furthermore, a negative test showed a NPV of 98.0% (*p*-value < 0.001), which is in line with previous studies [27]. Furthermore, we observed a large difference regarding the OR for FIT in the 2016–2018 and the 2019–2020 cohorts, with no apparent reason since the baseline data and reported FIT are similar. 

In order to assess the performance of the updated 2022 CRC fast-track criteria, we analyzed the pooled data. As expected, the new combination, comprising positive FIT combined with altered bowel habits in a patient > 40 years, was better at predicting CRC risk than altered bowel habits alone. Although we had a high number of missing cases for FIT since it was not previously part of the CRC fast-track criteria, it is clear that the use of a FIT increases the accuracy of any analysis with respect to predicting CRC risk. Interestingly, Swedish guidelines differ on the importance of FIT; it is not obligatory in the SCC-CRC pathway, while at the same time, a national screening program for CRC, solely based on FIT testing, has been implemented in Sweden, and in many other countries. 

There are some strengths in our study. Firstly, it contains all patients examined by colonoscopy according to the SCC-CRC during a 4.5-year period in our region. All patients were analyzed individually and in detail. The long time period enabled us to analyze two of three versions (2016 version and the 2019 update) of the fast track, and to make predictions for the 2022 update. The study populations were similar in baseline data and reported entry criteria, thus enabling a pooled analysis which strengthens the conclusions. We controlled for observer bias by reviewing the referrals before colonoscopy outcome. Inevitably, there are some limitations. Firstly, we assumed that symptoms not mentioned in the referrals were absent. Secondly, since we only included CRC diagnosed according to the SCC-CRC, patients diagnosed through other pathways were not included. We previously demonstrated that only 47.3% of all diagnosed CRC in our region were identified via SCC-CRC. In addition, 85% of those diagnosed via routine pathways were found to meet one or more SCC-CRC criteria, most frequently bleeding anemia [13]. 

There are many similarities within the European CRC fast-track pathways. However, most criteria are either poor predictors of CRC or are not clearly defined, leading to uncertainty regarding their interpretation and resulting in low cancer yields. Other factors not included in CRC fast tracks, such as abdominal pain, weight loss, smoking, inflammatory bowel disease, hereditary factors, and obesity, may also be associated with CRC risk. Finally, our results suggest that FIT testing should have a place in CRC fast-track pathways. An adequate filtering of referrals by a gastroenterologist seems to improve the cancer yield.

## 5. Conclusions

CRC yield was higher in Sweden, compared to other European countries, despite similar entry criteria, probably due to the assessment of the referrals by a gastroenterologist. The predictive values of the SCC-CRC criteria are generally poor; a significantly increased risk for CRC was only observed for abnormal radiology, abnormal rectal examination, anemia, and the now-removed entry criteria of changed bowel habits combined with blood in the stool. Of these, only abnormal radiology and abnormal rectal finding showed an OR >3. Testing for FIT, however, showed a high OR for CRC but also a NPV of almost 100%. We suggest that FIT should always be included in CRC fast-track pathways to increase diagnostic efficacy. 

We suggest the following entry criteria for CRC fast track: 

Patients > 40 years with one or more of the following signs/symptoms: 

- Positive FIT;

- Abnormal radiology;

- Abnormal rectal examination;

- Visible blood in stool, in the absence of hemorrhoids, as confirmed by rectoscopy.

## Figures and Tables

**Figure 1 cancers-15-04778-f001:**
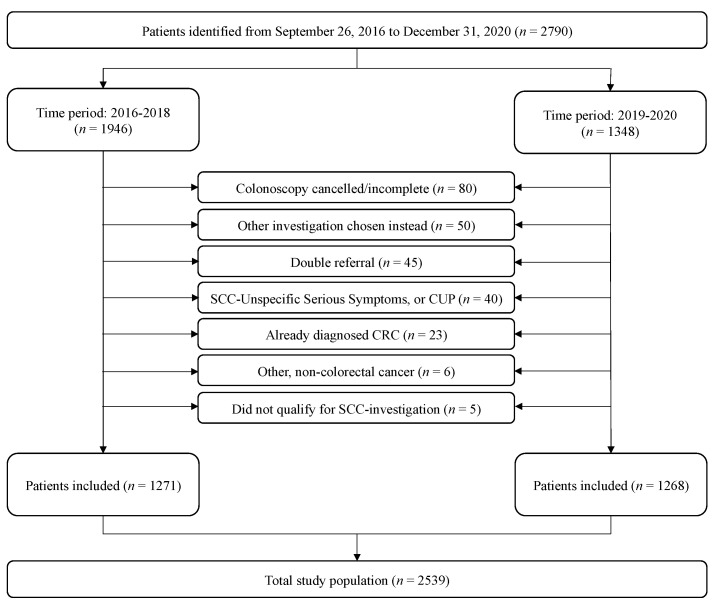
Flowchart of patient inclusion. In total, 251 patients were excluded from the study and the final study population included 2539 patients. The study population consisted of two groups: before and after the 2019 revision of the SCC-CRC entry criteria. Reasons for exclusion are displayed in the figure. The exclusion of cancelled/incomplete colonoscopies included patients that died before investigation, cancellation from either the patient or doctor, and incomplete investigation. Patients that did not qualify for SCC investigation had either a follow-up colonoscopy, or the investigation was wrongly prioritized as SCC-CRC. CUP = cancer of unknown primary origin.

**Figure 2 cancers-15-04778-f002:**
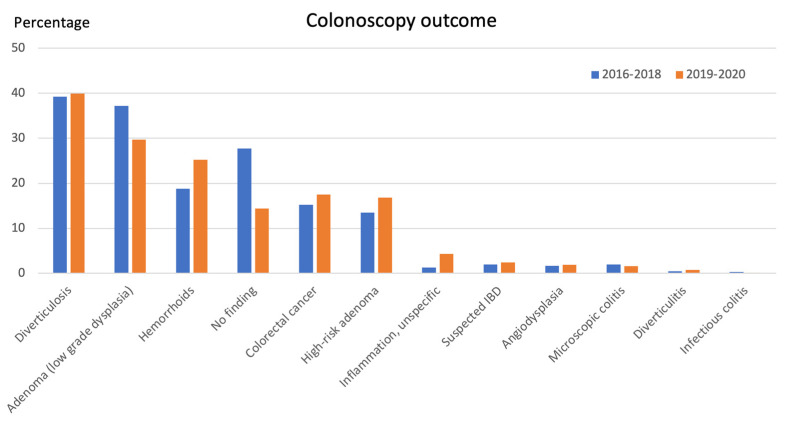
Findings from 2539 SCC-CRC colonoscopies in order of most prevalent to least prevalent finding. Findings are presented in percentages. High-risk adenoma = high-grade dysplasia, or adenomas ≥ 10 mm in size, or multiple adenomas (≥3), or villous histology, or serrated polyps with dysplasia or ≥10 mm in size. IBD = inflammatory bowel disease.

**Table 1 cancers-15-04778-t001:** Entry criteria for fast-track pathways for CRC assessment in different European countries.

Symptom/Sign	Sweden	Norway	Denmark	UK	Italy	Catalonia (Spain)	Valencia (Spain)
**Visible blood in stool/rectal bleeding**	•	•	•	•	•	•	•
**Anemia**	•		•	•	•	•	•
**Altered bowel habits**	• a)	•	•	•		•	•
**Finding from rectal examination**	•	•		• b)		•	•
**Weight loss**			• c)	•	•	•	
**Abdominal pain**			•	•			
**Radiological finding**	•				•		
**FOBT/FIT**				•			•
**Palpable abdominal mass**							•
**Other**	•		•	•	•	•	•
**Maximum days until endoscopy**	10 days	21 days	10 (14) days	2 weeks	10 days		Not specified

a) if combined with anemia/rectal bleeding, b) consider a 2WW with rectal mass, c) after individual consultation.

**Table 2 cancers-15-04778-t002:** Baseline characteristics of the study populations and for the pooled data, and associations to CRC diagnosis.

**Study Population 2016–2018 (*n* = 1271)**		**Cancer (*n* = 194)**	**No Cancer** **(*n* = 1077)**	** *p* ** **-Value**
	Age, median (IQR)	72.5 (13)	70 (19)	<0.001 *
	Male, *n* (%)	101 (52.1)	489 (45.4)	0.089 **
**Study population 2019–2020 (*n* = 1268)**		**Cancer (*n* = 222)**	**No cancer** **(*n* = 1046)**	
	Age, median (IQR)	73 (14)	70 (17)	<0.001 *
	Male, *n* (%)	120 (54.1)	467 (44.6)	0.011 **
**Study population pooled (*n* = 2539)**		**Cancer (*n* = 416)**	**No cancer** **(*n* = 2123)**	
	Age, median (IQR)	73 (13)	70 (17)	<0.001 *
	Male, *n* (%)	221 (53.1)	956 (45.0)	0.002 **

* = Analyzed by Mann–Whitney U test, ** = Analyzed by Pearson Chi^2^ test.

**Table 3 cancers-15-04778-t003:** Symptoms and findings and their respective predictive values.

SCC-CRC Entry Criteria	*n* (%)	*n* (%)	PPV (%)	NPV (%)	Specificity (%)	Sensitivity (%)	OR (95% CI)	*p*-Value
**SCC-CRC criteria** **2016–2018 (*n* = 1271)**	**Cancer ** **(*n* = 194)**	**No cancer** **(*n* = 1077)**						
Visible blood in stool/rectal bleeding	33 (17.0)	227 (21.1)	12.7	84.1	78.9	17.0	0.8 (0.5–1.1)	0.196
Anemia ^(a)^	101 (54.9)	324 (35.7)	23.8	87.5	64.3	54.9	2.2 (1.6–3.0)	<0.001
Abnormal rectal finding (rectoscopy/rectal examination)	51 (26.3)	124 (11.5)	29.1	87.0	88.5	26.3	2.8 (1.9–4.0)	<0.001
Abnormal radiology	52 (26.8)	126 (11.7)	29.2	87.0	88.3	26.8	2.8 (1.9–4.0)	<0.001
Altered bowel habits ^(b)^	81 (41.8)	582 (54.0)	12.2	81.4	46.0	41.8	0.6 (0.4–0.8)	0.002
**SCC-CRC criteria** **2019–2020 (*n* = 1268)**	**Cancer (*n* = 222)**	**No Cancer** **(*n* = 1046)**						
Visible blood in stool/rectal bleeding	57 (25.7)	286 (27.3)	16.6	82.2	72.7	25.7	0.9 (0.7–1.3)	0.612
Anemia ^(c)^	122 (56.2)	351 (39.6)	25.8	84.9	60.4	56.2	2.0 (1.5–2.6)	<0.001
Abnormal rectal finding (rectoscopy/rectal examination	53 (23.9)	143 (13.7)	27.0	84.2	86.3	23.9	2.0 (1.4–2.8)	<0.001
Abnormal radiology	65 (29.3)	140 (13.4)	31.7	85.2	86.6	29.3	2.7 (1.9–3.8)	<0.001
Combination of altered bowel habits and visible blood in stool/rectal bleeding ^(d)^	36 (16.2)	138 (13.2)	20.7	83.0	86.8	16.2	1.3 (0.9–1.9)	0.234
Combination of altered bowel habits and anemia ^(d)^	50 (22.5)	127 (12.1)	28.2	84.2	87.9	22.5	2.1 (1.5–3.0)	<0.001
**SCC-CRC criteria** **2016–2020 (*n* = 2539)**	**Cancer (*n* = 416)**	**No Cancer ** **(*n* = 2123)**						
Visible blood in stool/rectal bleeding	90 (21.6)	513 (24.2)	14.9	83.2	75.8	21.9	0.9 (0.7–1.1)	0.268
Anemia ^(e)^	223 (55.6)	675 (37.6)	24.8	86.3	62.4	55.6	2.1 (1.7–2.6)	<0.001
Abnormal rectal finding (rectoscopy/rectal examination	104 (25.0)	267 (12.6)	28.0	85.6	87.4	25.0	2.3 (1.8–3.0)	<0.001
Abnormal radiology	117 (28.1)	266 (12.5)	30.5	86.1	87.5	28.1	2.7 (2.1–3.5)	<0.001
Altered bowel habits ^(b)^	187 (45.0)	1117 (52.6)	14.3	81.5	47.4	45.0	0.7 (0.6–0.9)	0.004
Combination of altered bowel habits and anemia ^(d)^	83 (20.0)	234 (11.0)	26.2	85.0	89.0	20.0	2.0 (1.5–2.7)	<0.001
Combination of altered bowel habits and visible blood in stool/rectal bleeding ^(d)^	53 (12.7)	224 (10.6)	19.1	84.0	89.4	12.7	1.2 (0.9–1.7)	0.190

(a) Bleeding anemia according to laboratory Hb. Missing cases excluded from analysis: (%) CRC/No CRC 5.2/15.8. (b) = part of the SCC-CRC entry criteria prior to the 2019 revision. (c) Bleeding anemia according to laboratory Hb. Missing cases excluded from analysis: (%) CRC/No CRC 2.3/15.3. (d) = part of the SCC-CRC entry criteria after the 2019 revision. Missing data were excluded from the analysis. (e) Bleeding anemia based on laboratory hemoglobin values; missing cases CRC/No CRC (%) = 3.6/15.5. PPV = positive predictive value. OR = odds ratio. Pearson Chi^2^ test was used.

**Table 4 cancers-15-04778-t004:** Regression analysis for odds ratio (OR), adjusted for age and sex.

**SCC-CRC Criteria 2016–2018 (*n* = 1271)**	**OR (95% CI)**	** *p* ** **-Value**
Visible blood in stool/rectal bleeding	1.6 (1.0–2.5)	0.066
Anemia	2.2 (1.5–3.1)	<0.001
Abnormal rectal finding (rectoscopy/rectal examination)	4.1 (2.6–6.3)	<0.001
Abnormal radiology	5.0 (3.2–8.0)	<0.001
Change in bowel habits	1.0 (0.7–1.4)	0.968
**SCC-CRC criteria 2019–2020 (*n* = 1268)**		
Visible blood in stool/rectal bleeding	0.9 (0.5–1.5)	0.643
Anemia	2.3 (1.6–3.4)	<0.001
Abnormal radiology	4.6 (3.1–6.9)	<0.001
Abnormal rectal finding (rectoscopy/rectal examination)	3.3 (2.2–5.0)	<0.001
Combination of change in bowel habits and anemia	1.1 (0.7–1.7)	0.812
Combination of change in bowel habits and blood in stool	2.5 (1.3–4.7)	0.008
**SCC-CRC criteria 2016–2020 (*n* = 2539)**		
Visible blood in stool/rectal bleeding	0.9 (0.6–1.4)	0.705
Anemia	2.1 (1.5–3.0)	<0.001
Abnormal radiology	4.6 (3.4–6.3)	<0.001
Abnormal rectal finding (rectoscopy/rectal examination)	3.6 (2.7–4.8)	<0.001
Change in bowel habits	0.9 (0.6–1.3)	0.479
Combination of change in bowel habits and anemia	1.1 (0.7–1.8)	0.669
Combination of change in bowel habits and visible blood in stool	2.5 (1.4–4.5)	0.001

**Table 5 cancers-15-04778-t005:** Laboratory values for patients with and without CRC.

			Missing Values, CRC/No CRC (%)	PPV (%)	NPV (%)	*p*-Values	OR (95% CI)
**SCC-CRC criteria** **2016–2018 (*n* = 1271)**	**Cancer (*n* = 194)**	**No cancer (*n* = 1077)**					
**Positive FIT, *n* (%)**	84 (98.8)	402 (68.0)	56.2/45.1	17.3	99.5	<0.001	39.5 (5.5–285.8)
**Hb, g/L, median (IQR)** **- Males** **- Females**	120 (32) 124 (29) 118 (30)	130 (26) 136 (27) 128 (23)	5.2/15.8 8/16.4 2.1/15.3	N/A	N/A	<0.001 <0.001 <0.001	N/A
**SCC-CRC criteria** **2019–2020 (*n* = 1268)**	**Cancer (*n* = 222)**	**No Cancer (*n* = 1046)**					
**Positive FIT, *n* (%)**	97 (94.2)	379 (70.8)	53.6/48.9	20.4	96.3	<0.001	6.7 (2.9–15.5)
**Hb, g/L, median (IQR)** **- Males** **- Females**	123 (32) 130 (31) 114 (32)	129 (31) 135 (34) 127 (29)	2.3/15.3 1.7/15.0 2.9/15.5	N/A	N/A	0.005 0.177 <0.001	N/A
**SCC-CRC criteria** **2016–2020 (*n* = 2539)**	**Cancer (*n* = 416)**	**No cancer (*n* = 2123)**					
**Positive FIT, *n* (%)**	181 (96.3)	781 (69.4)	54.8/47.0	18.8	98.0	<0.001	11.4 (5.3–24.6)
**Hb, g/L, median (IQR)** **- Males** **- Females**	122 (32) 128 (31) 116 (30)	130 (28) 135 (32) 128 (25)	3.6/15.5 4.5/15.7 2.6/15.4	N/A	N/A	<0.001 <0.001 <0.001	N/A

FIT = fecal Immunochemical test; Hb = hemoglobin; PPV = positive predictive value; NPV = negative predictive value; N/A = not applicable. Statistics: FIT analyzed by Pearson Chi^2^ test; Hb analyzed by Mann–Whitney U test and presented in median and interquartile range due to non-normal distribution.

## Data Availability

The data presented in this study are available on request from the corresponding author. The data are not publicly available due to patient privacy.

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
