# Peer review of "Colorectal Cancer Fast Tracks: Cancer Yield and the Predictive Value of Entry Criteria"

_cancers, 2023, doi:10.3390/cancers15194778_

Round 1
Reviewer 1 Report
I was diagnosed with stage 3c colon cancer in 1997. My treatment fortunately worked well but a concern is that a sigmoid scan in 1996 found nothing. The tumor should have been detectable with a proper colonoscopy in 1996. Colonoscopy is a good practice but needs to be conducted at the proper time as needed for any patient. This paper showed to me that FIT is reliable to predict that colonoscopy should be conducted or not. It should be used everywhere in Europe instead of the much less reliable current practices. I appreciate that the authors strongly recommended the use of FIT for all countries in Europe.
Author Response
Thank you very much for taking the time to review this manuscript. Obviously there were no issues that we have to address.
Reviewer 2 Report
Uebel and colleagues examined fast track pathways for colorectal cancer diagnosis in Sweden. Although authors provide an important information, the lack of context for those who are not familiar with the European system will find the paper to be difficult to digest. Nevertheless, authors provided an important information that is valuable to the readers.
Specific comments
Please provide more fundamental information on the "fast-track" mechanisms in various European countries in the introduction section.
Please describe SCC- CRC in details in the introduction section.
Figure 2 should be represented as ratio rather than total cases. Also 2016-2018 and 2019-2020 should be represented in separate colomns for better comparison.
The way the discussion is constructed in confusing. It may be easier to to focus on the main findings of SCC-CRC data first and subsequently discuss the major differences in the outcome with other European programs.
Author Response
We thank the reviewer for his time and effort to improve the manuscript.
1: Please provide more fundamental information on the "fast-track" mechanisms in various European countries in the introduction section.
Respons: In line 47-50 we have made this more clear now. The details of the various fast-tracks in different countries are displayed in Table 1.
2: Please describe SCC- CRC in details in the introduction section.
Respons: In lines 53-57 and in 93-102 we have now added more detailed information regarding the SCC-CRC.
3: Figure 2 should be represented as ratio rather than total cases. Also 2016-2018 and 2019-2020 should be represented in separate colomns for better comparison.
Respons: We have changed Figure 2, according to the suggestion of the reviewer (Page 6).
4: The way the discussion is constructed in confusing. It may be easier to to focus on the main findings of SCC-CRC data first and subsequently discuss the major differences in the outcome with other European programs
Respons: We have now discussed the findings of the SCC-CRC more in the beginning of the discussion, line 251-254, especially focussing on the high adenoma detection rate in our study, which demonstrates that high-quality colonoscopies were performed.
Reviewer 3 Report
Overall important study, seem well planned out, organized, and statistically rigorous for clinicians. Some of the terms used could be adjusted to be more clinically relevant and clinically detailed so that readers can have more takeaways. Here are my considerations to perhaps make the discussion richer.
1. Some of the criteria seems vague, perhaps give examples in methods, mostly to clarify for clinical readers
-abnormal radiology: means thickening? Means mass?
-abnormal rectal finding: means by proctoscopy or rectal exam?
-change in bowel habits: means urgency? Diarrhea? Constipation? Thinner stool?
-in the absence of hemorrhoid: how do patients know that? By prior endoscopy?
-if subcategories are available consider including in supplements.
2. Where do these criteria come from for these fast track?
-consider discussing in introduction more
-does family history play into criteria? Why or why not?
3. Figure 2 is best to include % if possible. Horizontal instead of vertical may look better if including %.
Diverticulosis is not considered clinically relevant pathology, but certainly CRC and high risk adenomas and IBD are important. Were there criteria for diverticulosis complications such as bleeding?
4. Any information on whether patients have prior endoscopy hx. How soon ago?
5. Any comparison to other fast track or high risk programs out there? Or perhaps to compare to family hx population or average risk population.
Author Response
We thank the reviewer for his/her time and effort to improve the manuscript.
- Some of the criteria seems vague, perhaps give examples in methods, mostly to clarify for clinical readers
-abnormal radiology: means thickening? Means mass?
-abnormal rectal finding: means by proctoscopy or rectal exam?
-change in bowel habits: means urgency? Diarrhea? Constipation? Thinner stool?
-in the absence of hemorrhoid: how do patients know that? By prior endoscopy?
-if subcategories are available consider including in supplements.
Respons:
-abnormal radiology: indeed this is wall thickening or a suspect mass. We have clarified this in line 96-97.
- Abnormal rectal finding: indeed this is abnormal rectal palpation or rectoscopy. We have clarified this in line 95-96
- change in bowel habit: indeed this comprises diarrhea or constipation. We have clarified this in line 100.
- absence of hemorrhoids. This should be confirmed with rectoscopy by the referring doctor. We have made this more clear now in line 366.
- subcategories. There are no subcategories.
2. Where do these criteria come from for these fast track?
-consider discussing in introduction more
-does family history play into criteria? Why or why not?
Respons: these criteria are based on the fact that certain symptoms are assumed to be associated with an increased risk for cancer. In Sweden these criteria were formulated by the regional cancer center by a dedicated working group. This group includes representatives from all parts of the patient's health care chain, such as general practitioners, surgeons, oncologists, contact nurses, pathologists, and radiologists, as well as one or more patient representatives. We have added some extra text in the introduction, line 53-57.
Other factors such as family history, but also smoking or obesity, are not a part of the fast tracks criteria. We discussed these aspect in line 346-348, where we also mentioned hereditary factors.
3. Figure 2 is best to include % if possible. Horizontal instead of vertical may look better if including %.
Respons: this was also mentioned by another reviewer. We have changed Figure 2 now, with percentages.
Diverticulosis is not considered clinically relevant pathology, but certainly CRC and high risk adenomas and IBD are important. Were there criteria for diverticulosis complications such as bleeding?
Respons: No, there are no criteria for bleeding diverticulosis. Bleeding from diverticulosis may lead in visible blood in the stool, which, in turn, is a criterion for fast-track colonoscopy. However, in contrast to bleeding hemorrhoids, which can be diagnosed by the referring GP, visible bleeding from diverticulosis is not very common and can only be diagnosed by a colonoscopy, or in rare cases with a massive bleeding, by angiography. Since diverticulosis bleeding seems to be not very relevant in the context of CRC fast-tracks, we choose to not further discuss this aspect.
4. Any information on whether patients have prior endoscopy hx. How soon ago?
Respons: this is a relevant question and highlight the importance of the screening of the referrals by a gastroenterologist. Typically, most colon cancers develop from a polyp and this process usually takes many years. If a patient fullfils the SCC-CRC criteria, but the gastroenterologist notes that this patient recently underwent a colonoscopy, the chance that this patient suddenly has developed a CRC is considered minimal, and he/she will, for this reason, not enter the SCC-CRC pathway, but will be placed on a routine waiting list instead. We have added some extra text; lines 265-267.
5. Any comparison to other fast track or high risk programs out there? Or perhaps to compare to family hx population or average risk population.
Respons: In Sweden, there are fast-track pathways for almost all cancer forms, such as lungcancer, breast cancer, gastric/oesophagus cancer, and so on. Each fast-track has its own entry criteria. To compare the outcomes of these fast-tracks in comparison to the CRC fast-track is not possible, since there are no data available regarding the efficacy and the predictive values of the different entry criteria for other fast-tracks. Our study is the first study that critically evaluates the efficacy of the CRC fast-track pathway, including the predictive value of the entry criteria. Since it is beyond the scope (and the authors knowledge) of this paper to discuss the fast-track pathways for other cancers, we choose not to discuss this. However, we have added som text regarding the fact that there are many more fast-track pathways in Sweden; line 44-46.
Reviewer 4 Report
thank you for allowing me to review this original article. the manuscript is well written; the objectives are clearly stated and the statistical methodology adapted.
in particular, family history of colorectal cancer, predisposing genetic diseases, personal history of chronic inflammatory bowel disease.
in conclusion, the authors mention the role of screening by immunological testing. however, this test is recommended for asymptomatic patients. isn't this contradictory with the criteria mentioned, in particular transit disorders and red blood discharges from the anus?
Author Response
thank you for allowing me to review this original article. the manuscript is well written; the objectives are clearly stated and the statistical methodology adapted.
in particular, family history of colorectal cancer, predisposing genetic diseases, personal history of chronic inflammatory bowel disease.
Respons: Thank you for your valuable comments!
The criteria for the SCC-RC were formulated by the Regional Cancer Center by a dedicated working group. This group includes representatives from all parts of the patient's health care chain, such as general practitioners, surgeons, oncologists, contact nurses, pathologists, and radiologists, as well as one or more patient representatives. They choose not to include genetic factors, IBD and other factors. We agree that these factors also may be taken into account when it comes to the assessment of risk for CRC. In line 53-57 we have now described who is responsible for the SCC-CRC criteria, and in line 346-348 we mentioned that there are other factors that may contribute to CRC risk and we added inflammatory bowel disease as well.
in conclusion, the authors mention the role of screening by immunological testing. however, this test is recommended for asymptomatic patients. isn't this contradictory with the criteria mentioned, in particular transit disorders and red blood discharges from the anus?
Indeed, the FIT test is used in national screening programs because of its sensitivity and its high (99%) negative predictive value. For this reason, we advocate for this test in all cases where the physician suspects, or wants to exclude the possibility of colorectal cancer, because FIT obviously has the highest positive predictive value of all signs and symptoms. We have highlighted this in the conclusions, line 358-360. In case of continuous visible rectal bleeding, you may indeed expect a positive FIT and then it mandatory to perform a rectoscopy to see if the patients only has hemorrhoids, or a bleeding mass in the rectum.